# Dynamic Interplay between Lower-Grade Glioma Instability and Brain Metaplasticity: Proposal of an Original Model to Guide the Therapeutic Strategy

**DOI:** 10.3390/cancers13194759

**Published:** 2021-09-23

**Authors:** Hugues Duffau

**Affiliations:** 1Department of Neurosurgery, Montpellier University Medical Center, 34295 Montpellier, France; h-duffau@chu-montpellier.fr; Tel.: +33-4-67-33-66-12; 2Institute of Functional Genomics, University of Montpellier, 34295 Montpellier, France

**Keywords:** brain metaplasticity, lower-grade gliomas, mutational landscape, neuroplasticity, quality of life, white matter connectivity

## Abstract

**Simple Summary:**

The behavior of diffuse lower-grade glioma (LGG) is changing over time, spontaneously, and in reaction to therapies. Due to genomic instability and clonal expansion, although LGG initially progresses slowly, the growth accelerates at the time of malignant transformation. Furthermore, its progression pattern may change by switching from a proliferative towards a more migratory profile. Along with glioma plasticity, the brain itself is constantly adapting to the tumor and treatment(s) thanks to reconfiguration within and between neural networks. The pattern of reallocation can also evolve, especially by shifting from perilesional to contrahemispheric functional reorganization: this reorientation of cerebral reshaping is related to metaplasticity. The interplay between LGG mutations and reactional connectomal rearrangement leads to perpetual modulations in the glioma–neural equilibrium, explaining the possible preservation of quality of life. An original model of these dynamic interactions across LGG plasticity and the brain metanetwork is proposed to guide a tailored step-by-step individualized management over years.

**Abstract:**

The behavior of lower-grade glioma (LGG) is changing over time, spontaneously, and in reaction to treatments. First, due to genomic instability and clonal expansion, although LGG progresses slowly during the early period of the disease, its growth velocity will accelerate when this tumor will transform to a higher grade of malignancy. Furthermore, its pattern of progression may change following therapy, e.g., by switching from a proliferative towards a more diffuse profile, in particular after surgical resection. In parallel to this plasticity of the neoplasm, the brain itself is constantly adapting to the tumor and possible treatment(s) thanks to reconfiguration within and between neural networks. Furthermore, the pattern of reallocation can also change, especially by switching from a perilesional to a contrahemispheric functional reorganization. Such a reorientation of mechanisms of cerebral reshaping, related to metaplasticity, consists of optimizing the efficiency of neural delocalization in order to allow functional compensation by adapting over time the profile of circuits redistribution to the behavioral modifications of the glioma. This interplay between LGG mutations and reactional connectomal instability leads to perpetual modulations in the glioma–neural equilibrium, both at ultrastructural and macroscopic levels, explaining the possible preservation of quality of life despite tumor progression. Here, an original model of these dynamic interactions across LGG plasticity and the brain metanetwork is proposed to guide a tailored step-by-step individualized therapeutic strategy over years. Integration of these new parameters, not yet considered in the current guidelines, might improve management of LGG patients.

## 1. Introduction

Diffuse lower-grade glioma (LGG) is a complex brain neoplasm, able to change its behavior over time, spontaneously, and in reaction to therapies [1]. Although this tumor usually grows slowly during the early period of the disease, with a velocity of tumor diameter expansion about 3–4 mm of mean diameter per year, including in incidentally discovered LGG, it will inescapably accelerate its progression when it will evolve to a higher grade of malignancy [2]. Such a malignant transformation, which seems to be related to an accumulation of genetic mutations and epigenetic changes, is a complicated process, still poorly understood despite recent advances in molecular biology [3,4,5]. Beyond IDH1 mutation, which represents the earliest genetic alteration in LGG, longitudinal analyses at relapse have evidenced a high mutational potential of this tumor, especially with possible clonal expansion and epigenetic reprogramming after deletion or amplification of mutant IDH1 [6]. Genomic instability may also be facilitated by therapeutic agents such as temozolomide, which could result in hypermutation [7,8]. Radiotherapy could also be a possible cause of genomic instability together with chemotherapy [9,10]. In addition, the pattern of progression can change after treatment, in particular following surgery, e.g., by switching from a proliferative towards a more diffuse profile. Indeed, a bulky LGG on a preoperative MRI might exhibit a migratory pattern at recurrence after resection, which may sometimes mimic an image of gliomatosis [11]. This neoplasm, which commonly has loco-regional extension for several years, may also suddenly shift to a leptomeningeal diffusion around the brain or even around the spinal cord [12]. These recent findings demonstrate that LGG is constantly changing, explaining the considerable spatiotemporal heterogeneity both at cellular and macroscopical scales, and make it difficult to standardize the therapeutic management [13,14].

In parallel to this instability of the neoplasm, the nervous system itself is permanently adapting to the glioma growth and its treatment(s) thanks to mechanisms of cerebral reconfiguration [15]. In fact, brain functions rely on dynamic interactions within and between neural networks, leading to a perpetual succession of new equilibrium states, which allows physiological neuroplasticity (such as learning) as well as postlesional reshaping [16]. In particular, when a slow-growing tumor such as LGG arises, this focal lesion induces not only locoregional but also remote redistribution within the whole brain circuitry [17]. For instance, in the event of insular LGG, the contralateral insula is capable of adapting both structurally (with a volume increase) and functionally (with an increase of its functional connectivity), before any treatment, enabling a neurological compensation [18]. These phenomena of connectomal reconfiguration explain why LGG patients exhibit no or only very mild symptoms for many years until the onset of the first seizure usually revealing the tumoral disease, even when it involves areas traditionally thought as “eloquent” (e.g., Broca’s or Wernicke’s areas) [19]. It is worth noting that the neural plastic potential is also correlated to the time course of the lesion, with less efficient mechanisms of neural reallocation if the neoplasm is growing faster [20]. Remarkably, this neuroplastic potential may open the door to extensive surgical resections while preserving the quality of life, thanks to the intraoperative identification of critical cortical hubs and subcortical white matter (WM) pathways, especially by means of functional mapping in awake patients [21,22]. Interestingly, longitudinal studies based upon perioperative functional neuroimaging evidenced various patterns of pre- and post-surgical neural reallocation, with possible changes in the balance between the recruitment of perilesional versus contrahemispheric homotopic areas—knowing that a larger tumor usually generates a more distributed (bilateral) pattern of reconfiguration [17].

Therefore, it is crucial to better understand the interplay between LGG changes and reactional connectomal instability, which results in constant modulations in the glioma–neural equilibrium, both at ultrastructural and macroscopic levels. In this state of mind, reciprocal interactions across glioma cells and neurons have been shown [23]. On one hand, neuronal activity plays a pivotal role in promoting glioma progression through electrochemical neurogliomal synaptic communications [24]. In return, tumoral cells can be integrated in neural networks and are able to modulate them [25]. This incorporation of glioma cells into cerebral circuits may especially enhance cortical excitability in the infiltrated brain [26], resulting first in some degrees of neural reconfiguration and ultimately in epilepsy. In fact, seizures occur when the neuroplastic potential is overwhelmed, as supported by computational modeling, which took into account tumoral involvement of the WM tracts [27]. In this model, glioma-induced structural modifications were computed such that various aspects of the connectivity were damaged, mimicking the biological heterogeneity of LGG. Simulation demonstrated that tumor density changed the optimal neuroplastic regime, with performance loss in the neural circuit and eventually epilepsy in the case of dense WM invasion by the glioma [27]. These findings are in agreement with recent atlases of neuroplasticity, which evidenced that despite a high potential of cortical rearrangement, the main limitation of neural reshaping is represented by the WM connectivity [22,28]. For instance, by using diffusion imaging, a decrease in fractional anisotropy due to invasion of the right superior longitudinal fasciculus has been correlated with disturbances of visuospatial capacities [29], and changes in the microarchitecture of the left inferior fronto-occipital fasciculus have been correlated with semantic [30] and attentional impairments [31]. Thus, there is a progressive gradient of functional decline as the glioma gradually progresses, with several stages of aggravation: (i) initially, mild cognitive alterations are demonstrated on accurate neuropsychological assessment (possibly resulting in subjective complaints); (ii) then seizures; (iii) more severe cognitive disorders with ecological consequences on quality of life (e.g., for working); (iv) finally, disabling neurological deficits.

It has recently been suggested to redirect the pattern of neural reallocation [32]. Such a potential is related to meta-plasticity (plasticity of the synaptic plasticity), namely a higher-order plastic mechanism, which regulates the learning rule as a function of the dynamical context [33]. Indeed, in addition to the Hebbian plasticity, based upon the seminal idea that neurons firing together should wire together, and which generates rapid changes of the strength of specific synapses [34], non-Hebbian modifications of the synaptic activity may also arise over longer timescales, implying homeostatic plasticity and metaplasticity. The former modulates the overall activity level of neural circuitry while maintaining the relative strengths of synapses [35]; the latter changes the threshold needed to induce long-term potentiation depending on the past neural activity [32]. Such a regulatory plasticity interplays with Hebbian plasticity in order to prevent synaptic saturation and to preserve the equilibrium in the neural system [35,36]. This capacity for the plasticity itself to slowly adapt at the cellular level can have an impact on behavioral plasticity by learning to learn [37]. Interestingly, using metaplasticity might be helpful to reorientate mechanisms of cerebral reorganization in LGG patients [32]. In practice, it could be considered to generate a switch from a peritumoral rearrangement to a functional reshaping away from the glioma, with compensatory recruitment of remote ipsilesional and/or contrahemispheric structures, notably by means of cognitive rehabilitation and/or transcranial stimulation [38]. The ultimate aim would be to maximize the efficacy of neural delocalization, by adapting the profile of circuit redistribution to the behavioral modifications of the glioma, in order to maintain functional compensation over time.

## 2. Proposal of an Original Model Based on Dynamic LGG–Connectome Interplay

Such an improved knowledge of glioma–neurons dialogue, which depends on the LGG course and its changes due to genomic instability, as well as the real-time connectomal adaption, may enable to better predict the consequences of each therapeutic step and then to elaborate an optimal personalized management. To this end, an original model based on perpetual interactions between glioma plasticity and brain metaplasticity is proposed (Figure 1).

This 3DtM model takes account of several parameters (Figure 2):

The glioma spatial location and progression within the 3D architecture of the brain, on the basis of the anatomic sites involved (gyri/lobe(s); cortex/subcortical structures), the prominent focal growth versus prominent anisotropic migration along WM tracts (projection/associative fibers and/or commissural fibers—with a risk of bilateral diffusion). Of note, the index of neuroplasticity is not homogeneous within the connectome, thus defining a “minimal common brain”, i.e., a common structure across individuals, with a very low variability and a very low potential of functional compensation in the event of damage [39]. Furthermore, probabilistic atlases of plastic potential have already been computed based upon intraoperative electrostimulation mapping and functional-guided resection for LGG [22,28], and may be valuable to refine the individual “3D score” in the model—both for detailing glioma location and migration as well as for redefining structures at risk regarding surgery and radiotherapy [40]. In addition, repeat diffusion-weighted imaging could be helpful to quantify the invasion of the white matter tracts longitudinally [41].

The LGG velocity, correlated to the kinetics of neural networks reconfiguration reactional to the time constraint imposed by the tumor (t); to this end, a volumetric assessment of the glioma should be systematically achieved on the FLAIR-weighted imaging, with calculation of the mean tumor diameter (2 × volume)1/3, allowing us to compute the velocity diameter expansion (slope of the mean tumor diameter growth curve) by measuring the evolution of glioma diameter over time [2]. Since an average slope of 4.1 mm per year has been calculated in LGG [42], if a significant growth is demonstrated on two MRIs spaced 3 months apart (≥2 mm of mean diameter in 3 months, so ≥8 mm of mean diameter per year), this means that there is an acceleration of the growth rate, thus raising the question of a possible change in the management of patient [13].

In adaption to the behavioral modifications of the LGG over time (mutational instability and clonal expansion correlated to risk of malignant transformation, with acceleration of the growth rate and/or redirection to a more diffuse pattern of invasion), the metaplastic potential of the connectome with spontaneous and induced spatiotemporal reorientation of mechanisms of neural reorganization (M).

Thanks to this new model, the surgical and medical neurooncologists may anticipate more accurately the next treatment by tailoring the strategy over years based on the prediction pattern of LGG (re)growth and neural reallocation. Indeed, it is worth noting that none of these critical variables (pattern of 3D glioma migration within the brain with possible changes in the diffusion profile, tumor kinetics with possible growth acceleration, cerebral reshaping with possible metaplastic redirection of the neural system reconfiguration) have been taken into consideration in the current guidelines for LGG management [43].

## 3. Clinical Implications of the Model: Towards New Insights into the Oncofunctional Balance

In practice, exploring this constant glioma–brain interplay in a more systematic way could have major clinical implications, both from a functional and oncological perspective. First, in a very early stage of the disease, i.e., in a tumor with a low mutational burden and a slow growth (variable t positive [+]), usually before wide diffusion (variable 3D+), typically in incidentally discovered LGG [44], an objective evaluation of the cognitive functions should nonetheless be done and include an extensive neuropsychological assessment before to claim that the patient is “asymptomatic”. Importantly, a deficit was found in 60% of cases, in particular with an impairment of executive functions in 53% of patients with incidental LGG [44]. These findings plead in favor of a more precocious active treatment, ideally before the onset of epilepsy, i.e., a time point which already constitutes the overcompensated stage of neuroplasticity [24]. In fact, earlier surgery in a more plastic brain can result not only in a maximization of the extent of resection (due to a smaller tumor) [44,45], but also in an increase in the likelihood of postoperative recovery, due to a higher potential of neural circuitry reorganization (variable M+)—as supported by over 97% of return to work following resection for patients with incidental LGG [45]. Therefore, the oncofunctional balance of therapy, especially maximal surgical resection, is very positive at this stage.

Conversely, in symptomatic LGG patients (i.e., with epilepsy and/or neurological impairment), a later operation enhances the risk of leaving a greater amount of residual tumor, especially if the glioma invades the subcortical connectivity (variable 3D-), which has a very low potential of neuroplasticity (variable M-) [28,46]. A larger volume of postoperative residue is negatively correlated to overall survival [47,48] as well as negatively correlated to the quality of functional compensation [28]. Indeed, postoperative residual lesion infiltration has been linked to the degree of disconnection of WM pathways, e.g., by showing a deterioration of theory of mind (empathy) related to the degree of tumor involvement of the right frontoparietal connectivity (arcuate fasciculus and cingulum) [49], or lexical retrieval worsening related to postsurgical residual glioma volume within the left inferior longitudinal fasciculus [50]. Another argument pleading in favor of earlier treatment is to prevent malignant transformation, not only for oncological purposes, since the initial glioma volume is correlated to the risk of degeneration and to survival [51], but also for functional reasons. As previously mentioned, the potential of neural redistribution is less in rapid-growing lesions (variable t-), explaining the higher rate of neurological disorders in high-grade glioma [20]. In other words, the oncofunctional balance of surgical resection is still positive in symptomatic patients, on the condition nonetheless that the glioma did not yet accelerate its growth rate (t+) too much, did not migrate too widely in WM tracts (3D+), and in a patient with only mild disturbances meaning that the potential of neural reorganization is still high (M+).

In case of LGG recurrence after a first surgery, the therapeutic strategy should be tailored according to the new glioma–connectome equilibrium. Typically, when the glioma behavior results in a slow relapse (t+) with a prominent proliferative pattern, especially within the cortical areas (3D+), mechanisms of neuroplasticity can be optimal; thus, they make a second or even a third operation possible, with an increase in the extent of resection while preserving redistributed neural networks, which were able to exhibit further degrees of reconfiguration in comparison to the first surgery (M+) due to the slow LGG regrowth [52,53]. On the other hand, if the glioma re-progresses more rapidly (because of a malignant transformation) (t-) and/or with a more diffuse pattern (with prominent migration within the WM tracts) (3D-), the potential of neural compensation is lower (M-); this makes subsequent surgery more difficult because of a higher risk to induce permanent deficit or to achieve a very partial resection with no or only low impact on the LGG course [13,14]. Therefore, in this less favorable glioma/neural equilibrium, with a negative oncofunctional balance of surgery, medical adjuvant therapy should be considered.

Similarly, concerning medical treatment(s), the limitations of neuroplasticity (M) must be taken into account to preserve the quality of life when elaborating a personalized management, especially by considering the subcortical connectivity (3D). In fact, if the LGG switched to an invasive profile along the WM bundles, even with a slow growth (t), radiotherapy may generate a delayed disconnection syndrome by irradiating axonal fibers, due to the risk of demyelination [9]. Recent investigations using diffusion imaging after radiation therapy have evidenced a possible decrease in fractional anisotropy, which reflects injury of the subcortical pathways and which has been correlated with a decline in cognitive functions, such as verbal fluency impairment [54]. As a consequence, a redefinition of “organs at risk” has been suggested, in addition to the structures traditionally preserved (e.g., the brainstem, visual tracts, pituitary gland, hippocampi), in order to take account of the structural and functional connectivity of the brain in the irradiation planning [40]. Thus, in the event of a very diffuse profile of LGG, upfront chemotherapy may be discussed on the basis of multimodal considerations, especially the molecular profile [55] as well as the invasiveness of the glioma [13,14]. In this spirit, in case of wide LGG migration within the subcortical fibers, it has been proposed to administrate neoadjuvant chemotherapy, i.e., before (re)operation, in order to induce a glioma shrinkage with a reduction in the infiltration of the WM tracts, to enable opening the window to a subsequent resection while preserving cognition [56].

Table 1 summarizes positive or negative 3DtM variables and the clinical therapeutic approach suggested for each stage.

## 4. Conclusions and Perspectives

Beyond reducing or at least stabilizing the LGG volume to decrease accumulation of mutations and to prevent malignant transformation, new drugs should also be developed with the main goal of avoiding migration of glioma cells along the WM tracts. Indeed, since axonal connectivity represents the major limitation of neuroplasticity, a rapid (t-) and wide diffusion within the subcortical pathways (3D-) enhances the likelihood of cognitive decline. In parallel, redirecting the pattern of circuit reconfiguration by potentiating the recruitment of remote regions (especially the contrahemispheric regions) rather than the peritumoral areas (M+), may facilitate repeat locoregional treatments, as reoperations (including for gliomas deemed inoperable in so-called “eloquent areas” according to a rigid localizationist view of neural processing rather than in a dynamic networking framework) [16] and (re)irradiation over years while preserving the long-term quality of life of LGG patients. In other words, in clinical routine, the ultimate aim is to predict how the multistage therapeutic sequence should efficiently interact with the constantly changing neoplasm–brain intercommunication, by adapting each treatment step-by-step in a given patient at this specific moment (and by modulating the strategy in the next moment) in order to limit glioma mutational burden while canalizing metaplasticity of the cerebral connectome. Integration of these original parameters, not yet considered in the current recommendations, might improve management of LGG patients.

## Figures and Tables

**Figure 1 cancers-13-04759-f001:**
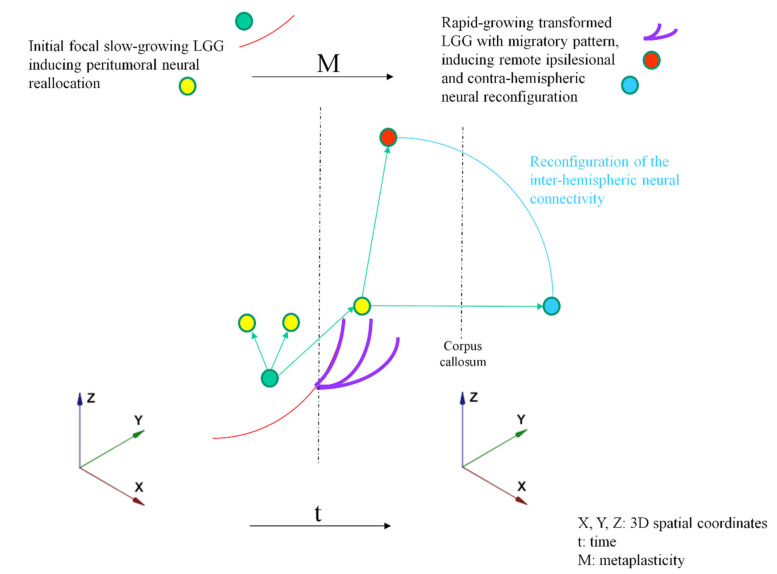
Graphical illustration of the 3DtM model. Tumor characteristics: Red line: Initial focal slow-growing LGG (first stage of the disease); Purple line: Rapid-growing transformed glioma with migratory pattern. Neural characteristics: Green arrow from green circle to yellow circle: peritumoral neural reallocation (in reaction to the first stage of the disease); Green arrow from yellow circle to red circle: recruitment of remote ipsilesional areas (in reaction to tumor progression); Green arrow from yellow circle to blue circle: recruitment of contra-hemispheric areas; Blue line: Reconfiguration of the interhemispheric neural connectivity remotely from the tumor.

**Figure 2 cancers-13-04759-f002:**
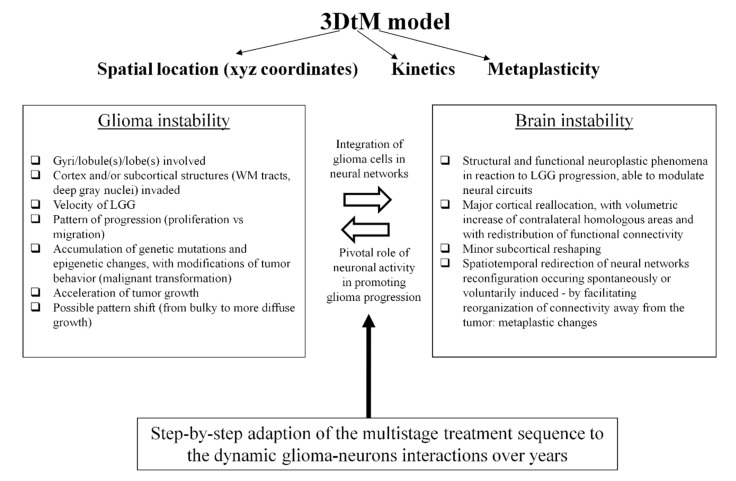
Modeling the continuously changing interplay between LGG progression and brain reactional reconfiguration to tailor the multistep therapeutic strategy.

**Table 1 cancers-13-04759-t001:** Clinical therapeutic approach based upon the 3DtM model.

Stage of the Disease	3DtM Model			Management
	Variable-3D	Variable-t	Variable-M	
Incidental discovery	+	+	+	Early maximal surgery
First seizure with normal cognitive assessment, limited invasion of WM, and slow LGG growth	+	+	+	Maximal surgery
Epilepsy with mild cognitive disorders related to moderate invasion of WM, and slow LGG growth	±	+	±	Surgery ± medical adjuvant treatment by privilegiating upfront chemotherapy
Epilepsy with significant cognitive disorders related to wide invasion of WM, and slow LGG growth	−	+	−	Neoadjuvant chemotherapy ± adjuvant surgery
Recurrent LGG with no or slight cognitive disorders, limited invasion of WM, and slow tumor growth	+	+	+	Reoperation
Recurrent glioma with no or slight cognitive disorders, limted invasion of WM, and acceleration of tumor growth rate	+	−	+	Reoperation followed by medical adjuvant treatments combining chemotherapy and radiotherapy
Recurrent LGG with moderate cognitive disorders despite wide invasion of WM but still slow tumor growth	−	+	±	Medical adjuvant treatment by privilegiating upfront chemotherapy ± reoperation
Recurrent LGG with significant cognitive/neurological deficits related to wide invasion of WM and/or acceleration of growth rate	−	−	−	Medical adjuvant treatments combining chemotherapy and radiotherapy

## Data Availability

No new data were created or analyzed in this study. Data sharing is not applicable to this article.

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
