# Peer review of "Dynamic Interplay between Lower-Grade Glioma Instability and Brain Metaplasticity: Proposal of an Original Model to Guide the Therapeutic Strategy"

_cancers, 2021, doi:10.3390/cancers13194759_

Round 1

Reviewer 1 Report

In this article the author presents a new model to guide therapeutic strategy in patients with lower grade gliomas based on three main variables: the spatial location and progression of gliomas (3D), tumour kinetics (t) and the metaplastic potential of the brain (M).

The aim of this article, which focuses on the dynamic gliomas-neuronal interaction, is of paramount importance from both neurosurgical and oncological perspectives. The article is well-written and the idea of a relatively simple model to guide the multidisciplinary management of these patients is very interesting.

I have, however, some comments and suggestions for the author.

1- In the introduction, I would suggest to include radiotherapy as well, as a possible cause of genomic instability together with chemotherapy as supported by several articles (Greene-Schloesser et al., 2012; Heeran et al., 2019, among others).

2- Regarding the 3DtM variables, I would appreciate a more detailed description:

  • Since there are many white matter bundles possibly invaded by LGG already at the incidental diagnosis, please suggest if the invasion of specific white matter bundles (for instance CST, AF or IFOF; specific portions? Side?)  would  determine a "3D-" score in the model. Applying a “minimal common brain” knowledge to the score may also have an impact on the timing and type of oncological treatment. It is, in fact, central to redefine the list of  "organs at risk" for radiotherapy (as mentioned later in the manuscript) according to specific white matter structures. This would also add some important spatial information merging location of the tumour and possible direction of invasion for longitudinal investigations.  

  • The author mentions at pages 5,6 and 7 a "wide migration" of tumour cells along the white matter fibres as a factor, possibly affecting the “3D” variable. I would suggest to provide more details regarding this aspect. A wider infiltration of white matter bundles represents a change in tumour invasiveness with important surgical and oncological implications. The measure of a "wide" white matter invasion is too generic and may result in a subjective interpretation of neuroradiological images with possible inaccuracies of the score model. Can the author try to provide a quantification of the tumour invasiveness over time to include in the score? 

  • Regarding the kinetics of LGG, It would help to add some details regarding the velocity of tumour growth /invasion.  Some estimations of volume changes between MRI investigations could guide the reader to understand, at what stage of volume increase a "t-" variable could be considered in the model. Does an interval of 3 or 6 months between investigation provide all the necessary information to change the score? What are the general recommendations regarding the time interval to detect changes in the score and possibly change the management of the patient (for instance reoperation)?

3- The Figure 1 is quite difficult to understand and in my opinion it does not illustrate the described variables. A 3D model or MRI images complemented with graph nodes or white matter reconstruction could help the author to better illustrate the model, increasing readability.

4- Since the overall aim of the article is to provide a guide for therapeutic strategy in clinical practice, I suggest to the author to create a table illustrating positive or negative 3DtM variables and the clinical-therapeutic approach suggested for each stage, in both primary and recurrent tumours. This would really increase the strength of recommendations and the overall impact of the score model based on the extensive experience of the author.

A table would also help to include the recommendation for neoadjuvant chemotherapy before surgery. Since this indication was not specifically included into the general EANO guidelines on the diagnosis and treatment of diffuse gliomas of adulthood (2021), this would clarify the author´s approach in these specific cases.

Author Response

In this article the author presents a new model to guide therapeutic strategy in patients with lower grade gliomas based on three main variables: the spatial location and progression of gliomas (3D), tumour kinetics (t) and the metaplastic potential of the brain (M).

The aim of this article, which focuses on the dynamic gliomas-neuronal interaction, is of paramount importance from both neurosurgical and oncological perspectives. The article is well-written and the idea of a relatively simple model to guide the multidisciplinary management of these patients is very interesting.

Author’s reponse: I thank the Reviewer for his/her positive comments.

I have, however, some comments and suggestions for the author.

1- In the introduction, I would suggest to include radiotherapy as well, as a possible cause of genomic instability together with chemotherapy as supported by several articles (Greene-Schloesser et al., 2012; Heeran et al., 2019, among others).

Author’s reponse: In the introduction, it is now mentioned that radiotherapy could be a possible cause of genomic instability together with chemotherapy (the references Greene-Schloesser et al., 2012 and Heeran et al., 2019 have been added).

2- Regarding the 3DtM variables, I would appreciate a more detailed description:

Since there are many white matter bundles possibly invaded by LGG already at the incidental diagnosis, please suggest if the invasion of specific white matter bundles (for instance CST, AF or IFOF; specific portions? Side?) would determine a "3D-" score in the model. Applying a “minimal common brain” knowledge to the score may also have an impact on the timing and type of oncological treatment. It is, in fact, central to redefine the list of "organs at risk" for radiotherapy (as mentioned later in the manuscript) according to specific white matter structures. This would also add some important spatial information merging location of the tumour and possible direction of invasion for longitudinal investigations.

Author’s reponse: In the model, in the part dedicated to the «3D structure-variable », it is now detailed that the index of neuroplaticity is not homogeneous within the connectome, thus defining a « minimal common brain », i.e. a common structure across individuals, with a very low variability and a very low potential of functional compensation in the event of damage (Ius et al., NeuroImage 2011). Furthermore, probabilistic atlases of platic potential have already been computed based upon intraoperative electrostimulation mapping and functional-guided resection for LGG (Herbet et al., Brain 2016 ; Sarubbo et al., NeuroImage 2020), and may be valuable to refine the individual « 3D score » in the model – both for detailing glioma location and migration as well as for redefining structures at risk regarding surgery and radiotherapy.

The author mentions at pages 5,6 and 7 a "wide migration" of tumour cells along the white matter fibres as a factor, possibly affecting the “3D” variable. I would suggest to provide more details regarding this aspect. A wider infiltration of white matter bundles represents a change in tumour invasiveness with important surgical and oncological implications. The measure of a "wide" white matter invasion is too generic and may result in a subjective interpretation of neuroradiological images with possible inaccuracies of the score model. Can the author try to provide a quantification of the tumour invasiveness over time to include in the score?

Author’s reponse: In the vein of the previous response regarding the involvement of the critical pathways by the glioma, it has been added that repeat diffusion-weighted imaging could be helpful to quantify the invasion of the white matter tracts longitudinally (Castellano et al. Evaluation of low-grade glioma structural changes after chemotherapy using DTI-based histogram analysis and functional diffusion maps. Eur Radiol 2016)

Regarding the kinetics of LGG, It would help to add some details regarding the velocity of tumour growth /invasion. Some estimations of volume changes between MRI investigations could guide the reader to understand, at what stage of volume increase a "t-" variable could be considered in the model. Does an interval of 3 or 6 months between investigation provide all the necessary information to change the score? What are the general recommendations regarding the time interval to detect changes in the score and possibly change the management of the patient (for instance reoperation)?

Author’s reponse: In the model, in the part dedicated to the « t-variable », it is now detailed that a volumetric assessment of the glioma should be systematically achieved on the FLAIR-weighted imaging, with calculation of the mean tumor diameter (2 × volume)1/3, allowing to compute the velocity diameter expansion (slope of the mean tumor diameter growth curve) by measuring the evolution of glioma diameter over time (Pallud et al., NeuroOncology 2013). Since an average slope of 4.1mm per year has been calculated in LGG (Mandonnet et al., Ann Neurol 2003), if a significant growth is demonstrated on 2 MRIs spaced 3 months apart (≥2 mm of mean diameter in 3 months, so ≥8 mm of mean diameter per year), this means that there is an acceleration of the growth rate, thus raising the question of a possible change in the management of patient (Duffau and Taillandier, NeuroOncology 2015).

3- The Figure 1 is quite difficult to understand and in my opinion it does not illustrate the described variables. A 3D model or MRI images complemented with graph nodes or white matter reconstruction could help the author to better illustrate the model, increasing readability.

Author’s reponse: As mentioned by the Reviewer « the idea of a relatively simple model to guide the multidisciplinary management of these patients is very interesting ». Therefore, the goal of this simple diagram is to illustrate schematically the recruitment of perilesional areas versus the recruitment of contrahemispheric neural networks in reaction to the glioma progression over time. In other words, the aim is not here to detail the complex functional anatomy of the brain (graph nodes or white matter reconstruction), knowing that comprehensive atlases of cortico-subcortical networks critical for the functions in LGG patients are now available (integrated in the Montreal Neurological Institute template) for the readers, as mentioned in the previous reply to the Reviewer (Sarubbo et al., NeuroImage 2020).

4- Since the overall aim of the article is to provide a guide for therapeutic strategy in clinical practice, I suggest to the author to create a table illustrating positive or negative 3DtM variables and the clinical-therapeutic approach suggested for each stage, in both primary and recurrent tumours. This would really increase the strength of recommendations and the overall impact of the score model based on the extensive experience of the author.

A table would also help to include the recommendation for neoadjuvant chemotherapy before surgery. Since this indication was not specifically included into the general EANO guidelines on the diagnosis and treatment of diffuse gliomas of adulthood (2021), this would clarify the author´s approach in these specific cases.

Author’s reponse: A table illustrating positive or negative 3DtM variables and the clinical-therapeutic approach suggested for each stage, in both primary and recurrent tumours, has been added.

Reviewer 2 Report

In this article, the author proposes a model to better guide the therapeutic management of patients with a lower-grade glioma (LGG). This model, termed 3DtM, considers the spatial location, the kinetics and the metaplasticity of lesions as three major parameters that shape the dynamic glioma-neurons interaction in the course of this disease. The manuscript is well-written and includes a novel concept that seems to have the potential of complementing and improving the current management strategies for patients with a LGG.

I do have the following minor concerns which I suggest to be addressed before considering publication:

p.1, Introduction, l.3: …..velocity of expansion of about…..

p.2, ll.4-5: please rephrase …..sometimes up to mimic a gliomatosis-like….. (this sentence seems unfinished)

p.2, ll.9-10: …..and make it difficult to standardize the therapeutic management. …..

p.2, l.13: Please include a reference.

p.2, l.53: …..diffusion imaging, a decrease…..

p.3, l.12: Please provide a more detailed explanation of Hebbian plasticity

p.5, 2nd paragraph, l.1: …..exploring this constant glioma-brain interplay in a more systematic way…..

p.5, 2nd paragraph, l.6: …..functions should be done and include an extensive neuropsychological…..

p.6, l.19: …..in rapid-growing lesions…..

Author Response

In this article, the author proposes a model to better guide the therapeutic management of patients with a lower-grade glioma (LGG). This model, termed 3DtM, considers the spatial location, the kinetics and the metaplasticity of lesions as three major parameters that shape the dynamic glioma-neurons interaction in the course of this disease. The manuscript is well-written and includes a novel concept that seems to have the potential of complementing and improving the current management strategies for patients with a LGG.

Author’s reponse: I thank the Reviewer for his/her positive comments

I do have the following minor concerns which I suggest to be addressed before considering publication:

p.1, Introduction, l.3: …..velocity of expansion of about…..

Author’s reponse: « velocity of expansion » has been replaced by « velocity of tumor diameter expansion » (in agreement with Pallud et al., 2013).

p.2, ll.4-5: please rephrase …..sometimes up to mimic a gliomatosis-like….. (this sentence seems unfinished)

Author’s reponse: this sentence has been rephrased as follows : « Indeed, a bulky LGG on preoperative MRI might exhibit a migratory pattern at recurrence after resection, which may sometimes mimic an image of gliomatosis-like »

p.2, ll.9-10: …..and make it difficult to standardize the therapeutic management. …..

Author’s reponse: corrected

p.2, l.13: Please include a reference.

Author’s reponse: the following reference has been added : « Duffau H. The huge plastic potential of adult brain and the role of connectomics: new insights provided by serial mappings in glioma surgery. Cortex. 2014 Sep;58:325-37»

p.2, l.53: …..diffusion imaging, a decrease…

Author’s reponse: corrected

p.3, l.12: Please provide a more detailed explanation of Hebbian plasticity

Author’s reponse: it is now mentioned that « the Hebbian plasticity is based upon the seminal idea that neurons firing together should wire together ». The following reference has been added : Hebb, D.O.(1949). The Organization of Behavior: A Neuropsychological Theory, Vol.44. New York, NY: Science Education.

p.5, 2nd paragraph, l.1: …..exploring this constant glioma-brain interplay in a more systematic way…..

Author’s reponse: corrected

p.5, 2nd paragraph, l.6: …..functions should be done and include an extensive neuropsychological…..

Author’s reponse: corrected

p.6, l.19: …..in rapid-growing lesions…..

Author’s reponse: corrected

Reviewer 3 Report

This article provide personal proposal of model of neural damage and reorganization, although no scientific evidence is provided.

Author Response

Author’s reponse: I thank the Reviewer for his/her comments. Further details have been provided in the new version, in response to the Reviewers 1 and 2.

Round 2

Reviewer 1 Report

I am thankful to the author for this revised version.  Almost all the comments and suggestions have been addressed. 

I have two further requests /comments:

1) Regarding the Figure 1, I understand the necessity to illustrate a simple and schematic model. However, the color choice and the different representations of white matter are not well explained. I suggest then, to implement the figure legend describing the colors and the forms displayed to increase readability of the figure and of the whole score model.

2) I appreciate the table summarising the general recommendations based on the model. However, I would recommend different mentions for radiotherapy and chemotherapy over time. It is of paramount importance to not induce genomic instability or myelin damage in young patients ultimately affecting the metaplastic potential when it is not strictly needed. I assume the author would leave the indication for radiotherapy when surgery is not possible anymore. In my opinion this is an important aspect, since the recent EANO guidelines include radiotherapy as a possible treatment already after the first surgery even in IDH mutated WHO grade 2 Astrocytomas.

Based on the extensive experience of the author, I would therefore, recommend to add, in the management column, a specific mention suggesting the best timing for radiotherapy or chemotherapy according to the score model.

Author Response

I am thankful to the author for this revised version. Almost all the comments and suggestions have been addressed.

I have two further requests /comments:

1) Regarding the Figure 1, I understand the necessity to illustrate a simple and schematic model. However, the color choice and the different representations of white matter are not well explained. I suggest then, to implement the figure legend describing the colors and the forms displayed to increase readability of the figure and of the whole score model.

Author’s reponse: The figure legend has been implemented, with more details regarding the colors and the forms displayed.

2) I appreciate the table summarising the general recommendations based on the model. However, I would recommend different mentions for radiotherapy and chemotherapy over time. It is of paramount importance to not induce genomic instability or myelin damage in young patients ultimately affecting the metaplastic potential when it is not strictly needed. I assume the author would leave the indication for radiotherapy when surgery is not possible anymore. In my opinion this is an important aspect, since the recent EANO guidelines include radiotherapy as a possible treatment already after the first surgery even in IDH mutated WHO grade 2 Astrocytomas.

Author’s reponse: The Table 1 has been completed, with more details regarding the indication for radiotherapy versus chemotherapy.

Round 3

Reviewer 1 Report

Thanks to the author for revising the manuscript according to my suggestions.